

**Insight into the in-cloud formation of oxalate based on in situ measurement**
**by single particle mass spectrometry**
Guohua Zhang[1], Qinhao Lin[1,2], Long Peng[1,2], Yuxiang Yang[1,2], Yuzhen Fu[1,2], Xinhui Bi[1,*], Mei
Li[3], Duohong Chen[4], Jianxin Chen[5], Zhang Cai[6], Xinming Wang[1], Ping'an Peng[1], Guoying
Sheng[1], Zhen Zhou[3]
[1] State Key Laboratory of Organic Geochemistry and Guangdong Key Laboratory of
Environmental Resources Utilization and Protection, Guangzhou Institute of Geochemistry,
Chinese Academy of Sciences, Guangzhou 510640, PR China
[2] University of Chinese Academy of Sciences, Beijing 100039, PR China
[3] Atmospheric Environment Institute of Safety and Pollution Control, Jinan University,
Guangzhou 510632, PR China
[4] State Environmental Protection Key Laboratory of Regional Air Quality Monitoring,
Guangdong Environmental Monitoring Center, Guangzhou 510308, PR China
[5] Shaoguan Environmental Monitoring Center, Shaoguan 512026, PR China
[6] John and Willie Leone Family Department of Energy and Mineral Engineering, The
Pennsylvania State University, University Park, PA 16802, USA
Correspondence should be addressed to Xinhui Bi (bixh@gig.ac.cn)



**Highlights**
● Single particle mixing state of oxalate in the cloud-free, residual, and interstitial particles
was first reported.
● Direct observational evidence showed the enhanced formation of oxalate in the cloud
residual and interstitial particles.
● Chemically segregated formation of oxalate was observed depending on the oxidized
organics associated with aged biomass burning particles.
● Glyoxylate served as an important intermediate for the formation of oxalate in the
troposphere of southern China.



**Abstract**
While ground-based works suggest the significance of in-cloud production (or aqueous
formation) to oxalate, direct evidence is rare. With the in situ measurements performed at a
remote mountain site (1690 m a.s.l.) in southern China, we first reported the size-resolved
mixing state of oxalate in the cloud droplet residual (cloud RES), the cloud interstitial (cloud
INT), and ambient (cloud-free) particles by single particle mass spectrometry. The results
support the growing evidence that in-cloud aqueous reactions promote the formation of oxalate,
with ~15% of the cloud RES and cloud INT particles containing oxalate, in contrast to only ~5%
of the cloud-free particles. Furthermore, individual particle analysis provides unique insight
into the formation and evolution of oxalate during in-cloud processing. Oxalate was
predominantly (>70% in number) internally mixed with the aged biomass burning particles,
highlighting the impact of biomass burning on the formation of oxalate. In contrast, oxalate was
underrepresented in aged elemental carbon particles, although they represented the largest
fraction of the detected particles. It can be interpreted by the individual particle mixing state
that the aged biomass burning particles contained an abundance of organic components serving
as precursors for oxalate. Through the analysis of the relationship between oxalate and organic
acid ions (-45$[HCO_2]^-$, -59$[CH_3CO_2]^-$, -71$[C_2H_3CO_2]^-$, -73$[C_2HO_3]^-$), the results show that in-
cloud aqueous reaction dramatically improved the conversion of organic acids to oxalate. The
abundance of glyoxylate associated with the aged biomass burning particles is the controlling
factor for the in-cloud production of oxalate. Since only limited information on oxalate is





available in the free troposphere, the results also provide an important reference for future
understanding of the abundance, evolution and climate impacts of oxalate.

**Keywords**: oxalate, individual particles, cloud droplet residues, mixing state, organic acids,
biomass burning



## 1 Introduction

In-cloud processing represents a large uncertainty in understanding the evolution and
impact of secondary organic aerosols (SOA) on both environment and climate (Ervens, 2015;
Ervens et al., 2011; Herrmann et al., 2015). Dicarboxylic acids significantly contribute to
SOA, aerosol acidity and hygroscopicity, and thus play an important role in atmospheric
chemistry and cloud condensation nuclei (CCN) (Ervens et al., 2011; Furukawa and
Takahashi, 2011; Sorooshian et al., 2013). Oxalic acid is globally the most abundant
dicarboxylic acid (Kawamura and Bikkina, 2016; Ho et al., 2010; Mochida et al., 2007),
accounting for as high as 5% of water soluble organic compounds downwind of the mainland
China (Feng et al., 2012; Kawamura and Bikkina, 2016). In addition, oxalate has great
impact on the solubility, photochemistry and bioavailability of transition metals in aerosols
(Ito and Shi, 2016; Johnson and Meskhidze, 2013).
Although there are primary sources, such as combustion of coal/biomass and biogenic
origins, oxalate is generally regarded as an oxidation product of malonate and glyoxylate,
precursors of which include glyoxal, methylglyoxal, glycolic acid, pyruvic acid, acetic acid
and so on (Carlton et al., 2006; Myriokefalitakis et al., 2011; Kawamura and Bikkina, 2016).
Large multifunctional compounds might also be important for the formation of oxalate
(Carlton et al., 2007). The formation pathways mainly include photochemical oxidation
followed by partitioning onto particulate phase and in-cloud aqueous formation (Yu et al.,
2005; Guo et al., 2016; Sullivan et al., 2007). The in-cloud aqueous pathway is generally
proposed as the dominant pathway based on the similar pattern between both size



distribution and concentration of oxalate and sulfate (Yu et al., 2005; Huang et al., 2006;
Laongsri and Harrison, 2013). However, Zhou et al. (2015) argued that only 16% of oxalate
could be attributed to in-cloud production, despite of its robust correlation with sulfate.
Photochemical oxidation could account for ~80% of oxalate in air mass influenced by
biomass burning (Kundu et al., 2010). More direct evidences are needed to better evaluate
the formation and behavior of oxalate during in-cloud processing. Through aircraft
measurements, Sorooshian et al. (2006) revealed higher concentration of oxalate in cloud
droplet residual (cloud RES) particles, rather than in cloud-free atmospheric particles over
Ohio, USA. Similarly, elevated oxalate levels due to in-cloud processing were observed
above coastal USA (Crahan et al., 2004), and Gulf of Mexico (Sorooshian et al., 2007a;
Wonaschuetz et al., 2012; Sorooshian et al., 2007b). Recently, an aircraft measurement also
provided an evidence on the important role of in-cloud production of oxalate from the near
surface to the lower free troposphere (i.e., ~2 km) over inland China (Zhang et al., 2016).
All of these in-situ observations were based on bulk particles analysis, and thus might miss
some valuable information on the mixing state of oxalate, which is demonstrated to be
significant for evaluating the life time and environmental impact of oxalate (Sullivan et al.,
2007; Zhou et al., 2015). Information on oxalate in the atmosphere associated with cloud
formation is still rare, far from enough for thoroughly understanding its distribution, sources,
formation, evolution and environmental impact (Kawamura et al., 2013; Meng et al., 2014;
Meng et al., 2013).



Single particle mass spectrometry (SPMS) has been commonly applied to obtain mixing
state of individual oxalate-containing particles, which is essential for their atmospheric
behaviors and environment impacts (Sullivan et al., 2007). Based on SPMS, oxalate was
found to be extensively internally mixed with sulfate in the Arctic boundary layer (Hara et
al., 2002). Similarly, the relative contributions of in-cloud processing, heterogeneous
reactions and biomass burning to oxalate in Shanghai was investigated (Yang et al., 2009).
Sullivan et al. (2007) demonstrated the significant contribution of photochemical formation
to oxalate followed by partitioning onto the dust and sea-salt particles. Zhou et al. (2015)
proposed that oxalate was readily photo-degraded in a form of oxalate-Fe complex in Hong
Kong. However, such studies have not been conducted to investigate the in-cloud formation
of oxalate. Investigation on the single particle mixing state of cloud/fog RES and interstitial
(cloud INT) particles would provide unique insight into the formation and aging processes
of aerosol compositions (Zhang et al., 2012; Bi et al., 2016; Li et al., 2011b; Pratt et al.,

2010).

To better understand the in-cloud aqueous formation of oxalate, we investigated
individual oxalate-containing particles at a high-altitude mountain site, representative of the
free troposphere in southern China. Using a single particle aerosol mass spectrometer
(SPAMS), the size-resolved mixing state of cloud-free, cloud RES and cloud INT oxalate-
containing particles were investigated. This paper reported data supporting the in-cloud
production of oxalate, and also discussed the influence of mixing state on the in-cloud
production.




**2 Methods**

**2.1 Cloud observation**



Measurements of the cloud-free, cloud RES, and cloud INT particles were performed at
the Nanling national background site (24°41′56″N, 112°53′56″E, 1690 m a.s.l.) in southern
China during 16-26 January 2016. Air masses from the southwestern continental and marine
areas dominated over the sampling period, bringing relatively warmer and wetter air masses
that benefited cloud formation (Lin et al., 2017), based on the back-trajectory analysis
(HYSPLIT 4.9, available at http://ready.arl.noaa.gov/HYSPLIT.php) by Air Resources Lab
(Draxler and Rolph, 2012). The air masses from northern areas, associated with cool dry
airstreams, arrived during 18 and 23-24 January, resulted in a decrease in both temperature
and relative humidity. Cloud events were characterized by a sudden drop in visibility (to <
5 km) and a sharp increase in relative humidity (> 95%) (Lin et al., 2017). In this study, three
long lasting (more than 12 hours) cloud events (Fig. 1), noted as cloud I, cloud II, and cloud
III, were identified. The visibility were generally lower than 1 km during the cloud events.
Aerosols were introduced into the instruments through two parallel sampling inlets. The
first one was a ground-based counterflow virtual impactor (GCVI) (Model 1205, Brechtel
Mfg. Inc., USA), applied to collect the cloud RES particles with a diameter greater than 8
μm. The GCVI employed a compact wind tunnel upstream of the CVI inlet (Model 1204)
to accelerate cloud droplets in the CVI inlet tip (Shingler et al., 2012). Upstream of the CVI
sampling tip, only droplets exceeding a certain controllable size (or cut size) could pass





through the counterflow and enter the evaporation chamber (with an air flow temperature at
40 °C), where the droplets were dried, leaving the cloud RES particles that are capable of
acting as CCN. A 15 L/min sample flow was provided to the downstream instruments. The
enhancement factor for particles concentration collected by GCVI was 5.25, corresponding
to the designation of the CVI. The detailed characterization and validation of the CVI
sampling efficiency could be found elsewhere (Shingler et al., 2012). The flow rates of the
whole GCVI system were validated before measurements, and were also automatically
monitored throughout the operation. A test on the cloud-free air showed that the average
particles number concentration sampled by the GCVI was ~1 cm$^{-3}$, in contrast to ~2000 cm$^{-}$
$^{3}$ in ambient air. The testing demonstrates that the influence of background particles on the
collection of the cloud RES particles could be negligible, further validating the performance
of the GCVI. In the present study, the average number concentration of the cloud RES
particles sampled during the cloud events was ~250 cm$^{-3}$ (Lin et al., 2017). The other one
ambient (PM$_{2.5}$) sampling inlet was used to deliver cloud-free or cloud INT particles.

**2.2 Instrumentation**

A SPAMS (Hexin Analytical Instrument Co., Ltd., Guangzhou, China), an

Aethalometer (AE-33, Magee Scientific Inc.), and a scanning mobility particle sizer (SMPS;
MSP Cooperation) were conducted to characterize the physical and chemical properties of
the sampled particles. During cloud I and cloud II, the instruments were connected
downstream the GCVI. During cloud III, cloud RES and cloud INT particles were alternately





sampled with an interval of ~1 h. During the cloud-free periods, these instruments were
connected to the ambient inlet in order to measure the cloud-free particles. The presented
results focused on the chemical composition and mixing state of the oxalate-containing
particles detected by the SPAMS. Therefore, details for other instruments were not provided
herein.

**2.3 Detection and classification of oxalate-containing particles**
The vacuum aerodynamic diameter ($d_{va}$) and mass spectral information for individual
particles could be obtained by the SPAMS (Li et al., 2011a). A brief description on
performance of the SPAMS can be found in the Supplement. Assuming Poisson distribution,
standard errors for the number fraction (Nf) of particles were estimated (Pratt et al., 2010),
since the particles were randomly detected by the SPAMS. Oxalate-containing particles are
identified as particles with ion peak at m/z -89 (Sullivan and Prather, 2007; Zauscher et al.,
2013), and their number-based mass spectra is shown in Fig. S1 in the Supplement.
Approximate 6000 particles were identified as oxalate-containing particles, accounting for
8.1 ± 0.1% of the total detected particles in the size range of 100-1600 nm. They were
clustered by an adaptive resonance theory-based neural network algorithm (ART-2a), based
on the presence and intensity of ion peaks (Song et al., 1999). Eight types with distinct mass
spectral characteristics (Fig. S2) were obtained for further analysis. More detail information
on all the observed particle types could be found elsewhere (Lin et al., 2017).



## 3 Results and Discussion

### 3.1 Direct observational evidence for in-cloud production of oxalate

The Nfs of the oxalate-containing particles relative to all the cloud-free, cloud RES, and cloud INT particles were $5.0 \pm 0.1\%$, $14.4 \pm 0.2\%$, and $13.4 \pm 1.1\%$, respectively (Table 1). The Nfs of the oxalate-containing particles varied from near zero in the cloud-free particles to ~80% in the cloud RES or cloud INT particles (Figure 1). Consistently, the average relative peak area (RPA) of oxalate in the cloud RES and cloud INT particles suppressed by a factor of ~8 that in the cloud-free particles. Defined as fractional peak area of each m/z relative to the sum of peak areas in a mass spectrum, RPA could represent the relative amount of a species on a particle (Jeong et al., 2011; Healy et al., 2013). At ground level in China, oxalate was found in ~3% of total particles in Shanghai (Yang et al., 2009) and the PRD region (Cheng et al., 2017), respectively. Relatively higher fraction of oxalate-containing particles in this study might reflect the importance of atmospheric ageing during long-range transport for the formation of oxalate at the high mountain site of southern China.

Analogous Nfs of the oxalate-containing particles in the cloud RES and cloud INT particles suggest the similar formation mechanism of oxalate in cloud droplets and interstitial particles, although Dall'Osto et al. (2009) indicated that difference might exist for secondary compounds formation between fog droplets and INT particles. The Nfs of the oxalate-containing particles in the cloud-free, cloud RES, and cloud INT particles versus $d_{va}$ are displayed in Fig. 2. Oxalate-containing particles had higher Nfs in the smaller cloud-free particles, indicative of primary emission or photochemical production followed by



condensation (Zauscher et al., 2013). On the contrary, the Nfs of the oxalate-containing
particles in the cloud RES and cloud INT particles increased with increasing $d_{va}$, showing a
distinctly different pattern. It indicates that in-cloud aqueous reaction grows the cloud RES
and cloud INT oxalate-containing particles with addition of secondary compositions
(Schroder et al., 2015). It is further supported by the unscaled number size distribution of
the cloud-free, cloud RES, and cloud INT oxalate-containing particles (Fig. S3), with $d_{va}$
peaking at around 0.5, 0.8, and 0.7 μm, respectively.
It is further shown that the enhanced Nfs of the oxalate-containing particles was not
likely due to the influence of air mass. Firstly, the Nfs of the cloud-free oxalate-containing
particles were generally low (< 10%) over the sampling period (Fig. 1 and Fig. S4), reflecting
a background level of oxalate. Secondly, the Nfs and the RPAs of the cloud RES oxalate-
containing particles exclusively sharply increased when RH was larger than 95% (Fig. S4).
Significant enrichment of oxalate in the cloud RES particles demonstrates the importance of
in-cloud aqueous reactions in the formation of oxalate (Sorooshian et al., 2006). Overall,
these results provide direct evidences that the in-cloud aqueous processing is the dominant
mechanism for oxalate in this study. More details on the formation mechanism and the
dominant influence factors would be discussed in the following text.

**3.2 Predominant contribution of biomass/biofuel burning to oxalate**
Number fractions of the major ion peaks associated with the oxalate-containing particles
were compared to those with all the detected particles, as shown in Fig. 3. Detailed



information on the Nfs of all the detected ion peaks in the oxalate-containing particles could
be found in Fig. S1. Potassium, with intense peak (peak area > 1000) at m/z 39  Da, is
ubiquitously (~90%) associated with the oxalate-containing particles. It is attributed to
highly sensitive of potassium to the desorption laser in the SPAMS, although m/z 39 Da may
also be appointed to $39[C_3H_3]^+$ (Silva et al., 1999). Sulfate ($-97[HSO_4]^-$, 96%) and nitrate ($-$
$62[HNO_3]^-$, 88%) were the dominant secondary inorganic species associated with the
oxalate-containing particles. Other major ion peaks were ammonium ($18[NH_4]^+$, 47%),
organic nitrogen ($-26[CN]^-$, 76%), and oxidized organics (i.e., m/z -45, -59, -71, and -73)
with the Nfs ranging from 17% to 57%. These oxidized organics were commonly found in
aged biomass burning particles, regarded as organic acids (OAs). Their RPAs increased with
increasing particle sizes (Fig. S5), indicative of secondary origins (Zauscher et al., 2013).
Furthermore, these OAs, most likely assigned to be formate at m/z $-45[HCO_2]^-$, acetate at
m/z $-59[CH_3CO_2]^-$, methylglyoxal or acrylate at m/z $-71[C_2H_3CO_2]^-$, and glyoxylate at m/z
$-73[C_2HO_3]^-$  (Zauscher et al., 2013), tracked each other temporally (Table S1), supporting
their similar formation mechanisms. Other OAs with minor fraction (~10%) were also
detected to be associated with the oxalate-containing particles, such as m/z -87, -103, and -
117 Da due to pyruvate, malonate, and succinate, respectively. OAs could be formed through
oxidation of volatile organic compounds in biomass burning plume (Zauscher et al., 2013).
Continuous evolution of primary organics to highly oxidized organics is widely observed for
biomass burning particles (Zhou et al., 2017; Cubison et al., 2011). The extensive presence
of potassium, OAs, and organic nitrogen in the oxalate-containing particles reflects the





substantial contribution of biomass burning to the observed oxalate (Zauscher et al., 2013;
Pratt et al., 2010). The oxalate-containing particles observed herein likely represented aged
biomass burning particles with enhanced aliphatic acids (Paglione et al., 2014). Significant
correlations between these OAs were observed in aged biomass burning particles (Zauscher
et al., 2013) and also cloud water samples (Sorooshian et al., 2013). Hence, it is expected
that the Nfs of these OAs were obviously larger in the oxalate-containing particles, rather
than those in the other detected particles (Fig. 3).

The contribution of biomass burning to the observed oxalate could also be reflected by

the overwhelming potassium-rich (K-rich) particles (Table 1 and Fig. S2), regarded as aged
biomass burning particles herein (Bi et al., 2011; Pratt et al., 2010; Zauscher et al., 2013).
Following emission, biomass burning particles become enriched in sulfate, nitrate, and OAs
as ageing processes (Reid et al., 2005). It can be seen in Fig. 4 that 75.1 ± 1.5% of oxalate
was associated with the K-rich particles, although they only accounted for 36.0 ± 0.3% of
all the detected particles (Lin et al., 2017). Only 4.0 ± 0.4% of oxalate was associated with
the aged elemental carbon (EC) particles although they were the dominant fraction (45.0 ±
0.3%) of all the detected particles, reflecting an external mixing state. Enhancement of
oxalate in the K-rich particles supports the favorable formation of oxalate in aged biomass
burning particles. Such a high fraction (i.e., 75.1 ± 1.5%) in the present study indicates a
substantial contribution from secondary processing of biomass burning particles, as
discussed above. The result is consistent with previous studies that observed abundance of
oxalate substantially influenced by aged biomass burning particles (Gao et al., 2003;



Deshmukh et al., 2016; Yang et al., 2014; Zhou et al., 2015). Primary emission from biomass
burning contributes only a minor fraction to the observed oxalate in the atmosphere in China
(Meng et al., 2013; Yang et al., 2009). Direct observation also supports the absence of
oxalate in primary biomass burning particles (Silva et al., 1999; Huo et al., 2016).

As shown in Fig. 4, ~10% of oxalate was associated with Fe-rich particles, second only

to the K-rich particles. Regarding that the Fe-rich particles only accounted for $2.5 \pm 0.4\%$ of
all the detected particles (Lin et al., 2017), it might reflect that the Fe facilitated the formation
of oxalate. Fenton reactions involving iron can produced more oxidants (e.g., •OH)
(Herrmann et al., 2015; Nguyen et al., 2013), which is an important factor for the formation
of oxalate (Ervens et al., 2014). Likewise, the highest fraction (> 30%) of oxalate was found
to be internally mixed with metal-containing (e.g., iron, zinc, copper) particles in the Pearl
River Delta region (Cheng et al., 2017). Oxalate was also found to be slightly enriched in
amine-containing particles, which is most probably attributed to the enhanced partition of
amine to wet aerosols (Zhang et al., 2012; Rehbein et al., 2011).

**3.3 Pathway for in-cloud formation of oxalate in aged biomass burning particles**

As shown in Table 1, > 70% of oxalate by number was associated with the aged biomass

burning particles. It is also noted that ~10% of the cloud-free K-rich particles contained
oxalate, while the fraction increased to > 20% in the cloud INT and cloud RES K-rich
particles. This is not likely due to the preferential activation of the K-rich particles, since the
Nfs of oxalate associated with the K-rich particles is similar (70-76%) for the cloud-free,



cloud RES, and cloud INT particles (Fig. S6). Therefore, the favorable formation of oxalate
in the K-rich particles is most probably attributed to the enhanced organic precursors, as
discussed in the following.

The major OAs were predominantly associated with the oxalate-containing particles

(Fig. 3) and also the K-rich particles (Table S2). Furthermore, significant correlations ($p <$
0.01) were found for the temporal profiles of the Nfs of the OAs and that of the oxalate-
containing particles, particularly, for the cloud RES particles (Table S1). The highest
correlation was found between the oxalate-containing particles and the glyoxylate-
containing particles in the Nf and the RPA (Fig. 5). The correlations were significantly
stronger for the cloud RES and cloud INT particles rather than for the cloud-free particles,
suggesting the in-cloud production from glyoxylate as an important pathway for oxalate. It
should further confirm the assignment of m/z -73 to glyoxylate, regarded as one of the
primary intermediates contributing to formation of oxalate (Carlton et al., 2006;
Myriokefalitakis et al., 2011). Miyazaki et al. (2009) suggested that secondary production of
oxalate probably in aqueous phase is important via the oxidation of both longer-chain diacids
and glyoxylate, and would be enhanced in biomass burning influenced particles. To our
knowledge, it is the first report on the direct link and the internally mixing state between
glyoxylate and oxalate during in-cloud processing with high time resolution. Additionally,
the linear regression slopes between glyoxylate and oxalate for the cloud RES and cloud INT
particles were also higher than that for the cloud-free particles (Fig. 5), which also supports
the more effective production of oxalate in cloud.





We further analyzed the relative fraction of the peaks areas of oxalate, glyoxylate, and
OAs in oxalate-containing particles during the cloud-free periods and cloud events (Fig. 6).
It can be seen that the dots distribute close to the OAs during cloud-free periods, whereas
they distribute towards oxalate during cloud events. This distribution indicates that the OAs
were the dominant composition relative to oxalate and glyoxylate in the cloud-free oxalate-
containing particles, whereas oxalate became more important in the cloud RES and cloud
INT oxalate-containing particles. The different pattern is attributable to the conversion of
the OAs to oxalate as a result of in-cloud aqueous reactions. It is also supported by the
variations of the Nfs of the major OAs in the cloud-free, cloud RES, and cloud INT particles,
respectively (Fig. S7). A substantial decrease (~50% on average) is found for the Nfs of the
OAs associated with the oxalate-containing particles, from the cloud-free particles to the
cloud RES and cloud INT particles. On the other hand, the Nfs of the OAs in all the detected
particles did not show an obvious decrease. The conversion of the OAs to oxalate during in-
cloud processing is consistent with the observation that oxalate increased as the droplets
evaporated, while acetate, glyoxylate, and malonate decreased (Sorooshian et al., 2007b).
Most of previous studies considered that glyoxylate is dominantly produced from
aqueous oxidation of glyoxal or glycolic acid, depending on volatile organic compounds
(Sorooshian et al., 2006; Sorooshian et al., 2007b; Ervens et al., 2004). Aqueous phase
reaction promotes the production of oxalate through increasing the partitioning of gases into
droplets (Sorooshian et al., 2007a). If this pathway dominated in this study, glyoxylate and
oxalate should be evenly distributed in all the particle types, which is inconsistent with our





observation that they were predominantly associated with the aged biomass burning particles
(Fig. 3). It indicates that a certain amount of glyoxylate should be directly produced in cloud
from the organics formed before the cloud events and associated with aged biomass burning
particles. Aqueous-phase processing of biomass-burning emissions was demonstrated to be
a substantial contributor to the SOA (Gilardoni et al., 2016). Existing models typically treat
cloud droplets as a well-mixed bulk aqueous phase (McNeill, 2015), and initialize the
particle composition as pure ammonium sulfate (Ervens et al., 2004; Sorooshian et al., 2006).
Our results suggest that a particle type based model with detailed chemical mixing state is
required for further understanding on the modification of particle properties by in-cloud
processing in the troposphere.

**3.4 Case study for the influence of air mass on the formation of oxalate**

Cloud II represented a relatively more polluted condition, with PM$_{2.5}$ around 200 ng m$^-$

$^3$, ~4 times those during cloud I and III. Air mass analysis showed that cloud II was strongly
influenced by northeastern air mass, contrasting to the southwestern air mass during cloud I
and III (Lin et al., 2017). Figure 7 compares the respective Nfs of the K-rich, oxalate-
containing, and glyoxylate-containing particles during the three cloud events. The K-rich
particles were found to contribute ~25% of the cloud RES particles during cloud II, which
was significantly lower than its contribution (~50%) during cloud I and III. Similarly, Nf of
the glyoxylate-containing particles during cloud II was significantly lower, which is also
similar for other oxidized organics (Table S3). Since oxalate was predominantly associated



with the aged biomass burning particles, Nf of the oxalate-containing particles shares a
similarly trend. This is because the in-cloud production of oxalate on the aged biomass
burning particles is dominantly controlled by the glyoxylate. It is also supported by higher
correlation between the Nfs of oxalate-containing and glyoxylate-containing particles,
relative to that between the Nfs of oxalate-containing particles and the aged biomass burning
particles (Table S1). The result suggests that aged biomass burning particles from
northeastern air mass contained less amount of oxidized organics for the formation of oxalate.
We also note that short cloud processing time should not be the reason for the lower Nf of
oxalate-containing particles during cloud II. As can be seen in Fig. 1, the Nf of oxalate-
containing particles increased to 20% within several hours during cloud I and III.

**4 Conclusions**
Individual particle mixing state of oxalate in the cloud-free, cloud RES and cloud INT
particles obtained at a remote mountain site allows for the investigation of formation and
evolution of oxalate. Our results show significant enhancement of oxalate-containing
particles in the cloud RES and cloud INT particles, rather than in the cloud-free particles,
providing first direct observational evidence for the in-cloud production of oxalate in the
troposphere in China, and strengthening the growing evidence that aqueous-phase chemistry
is the predominant formation mechanism for oxalate. The influence of biomass burning on
the formation of oxalate was also highlighted, with predominant fraction (> 70%) of oxalate
internally mixed with aged biomass burning particles. Formation of oxalate is highly



dependent on the abundance of organic acids strongly associated with the aged biomass
burning particles, with glyoxylate as an important intermediate. In-cloud chemically
segregated production of oxalate would lead to a substantial change of the biomass burning
particles after cloud evaporation, different from other particle types (e.g., aged EC particles
externally mixed with oxalate). It would have important implication for accurate modeling
the formation and influence of oxalate in the atmosphere.

**Acknowledgement**
This work was supported by the National Key Research and Development Program of
China (2017YFC0210104), the National Nature Science Foundation of China (No.
91544101), and the Foundation for Leading Talents of the Guangdong Province Government.
G.H. Zhang would like to thank the support from State Key Laboratory of Organic
Geochemistry (SKLOG2016-A05).



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





**Tables**
**Table 1. The number and number fraction of oxalate-containing particles in**
**the all the detected cloud-free, RES, and INT particles.**

|  | Cloud-free | Cloud RES | Cloud INT |
|---|---|---|---|
| Num. of all the detected particles | 48835 | 23616 | 1063 |
| Num. of oxalate-containing particles | 2442 | 3410 | 142 |
| Nf. of oxalate-containing particles | 5.0 ± 0.1% | 14.4 ± 0.2% | 13.4 ± 1.1% |
| Nf. of oxalate-containing particles classified as aged biomass burning particles | 76.3 ± 1.8% | 70.0 ± 1.4% | 71.8 ± 7.1% |




**Figure caption**
Fig. 1. (a) Temporal variation (in one-hour resolution) of Nfs of the oxalate-
containing particles, and box-and-whisker plots of (b) the Nf of oxalate-containing
particles, and (c) the relative peak area (RPA) of oxalate, separated for the cloud-free,
cloud RES, and cloud INT particles. In a box and whisker plot, the lower, median and
upper line of the box denote the 25, 50, and 75 percentiles, respectively; the lower and
upper edges of the whisker denote the 10 and 90 percentiles, respectively. Red triangles
refer to the arithmetical mean values of the Nfs and RPAs shown in (b) and (c).
Fig. 2. Size dependent Nfs of oxalate-containing particles relative to all the
detected cloud-free, cloud RES, and cloud INT particles, respectively.
Fig. 3. Number fractions of the major ion peaks in oxalate-containing and all the
detected particles, respectively.
Fig. 4. Number fractions of the single particle types for oxalate-containing and all
the detected particles, respectively.
Fig. 5. Correlation analysis between (a) the Nfs and (b) The RPAs of the oxalate-
containing and glyoxylate-containing particles, separated for the cloud-free, cloud RES,
and cloud INT particles, respectively.
Fig. 6. The relative distributions of the peak areas of oxalate, glyoxylate, and the
OAs for (a) the individual cloud-free and (b) the cloud RES and cloud INT oxalate-
containing particles. The peak areas of the OAs were summed from those of the
individual OAs. The coloration indicates the RPA of oxalate.



Fig. 7. Box and whisker plots of the variations of Nfs for the K-rich, oxalate-

containing, and glyoxylate-containing particles during the cloud events, respectively.



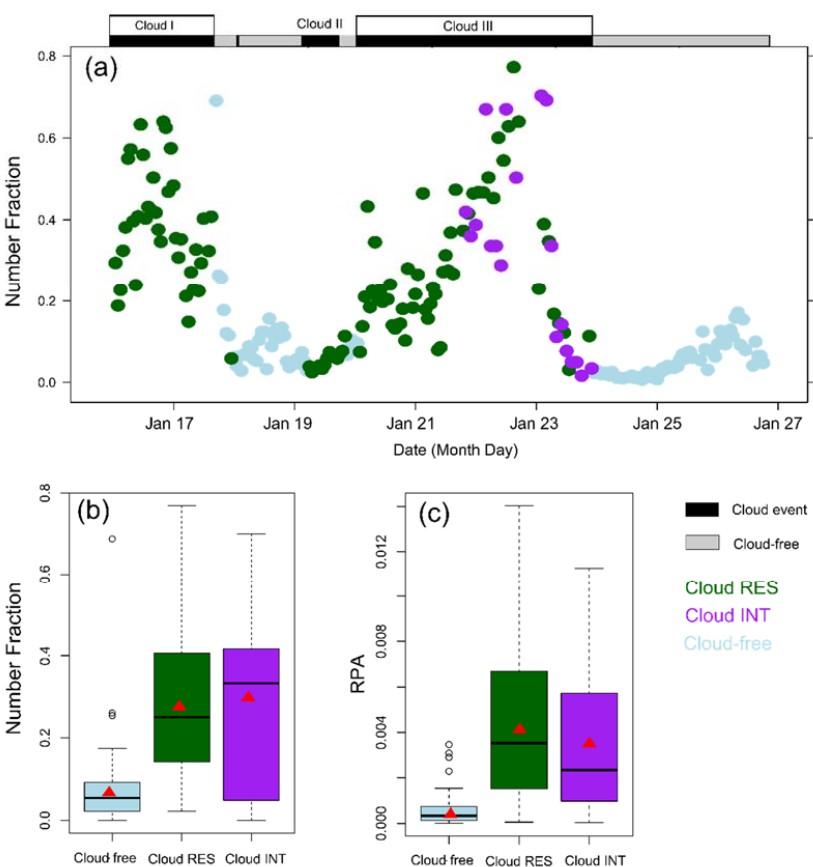


Fig. 1.





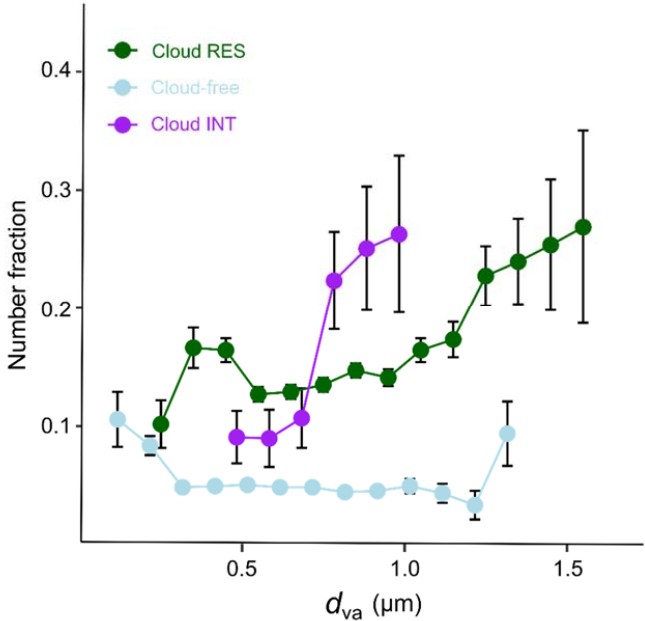


Fig. 2.





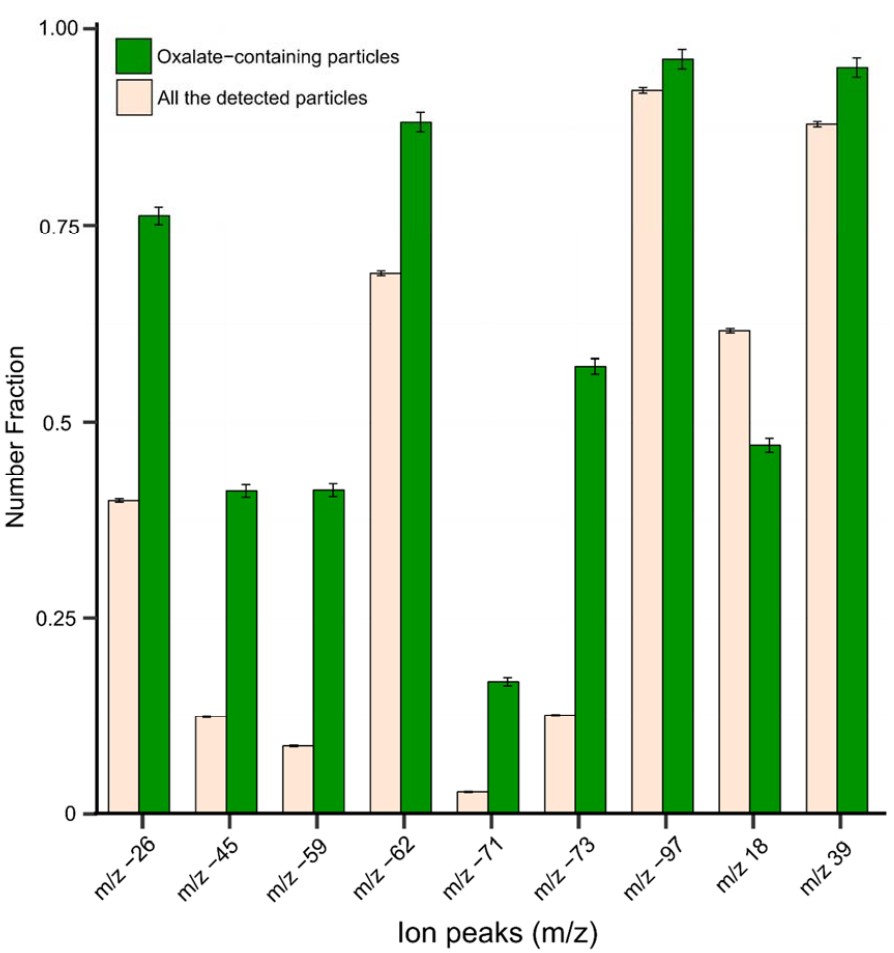


**Fig. 3.**





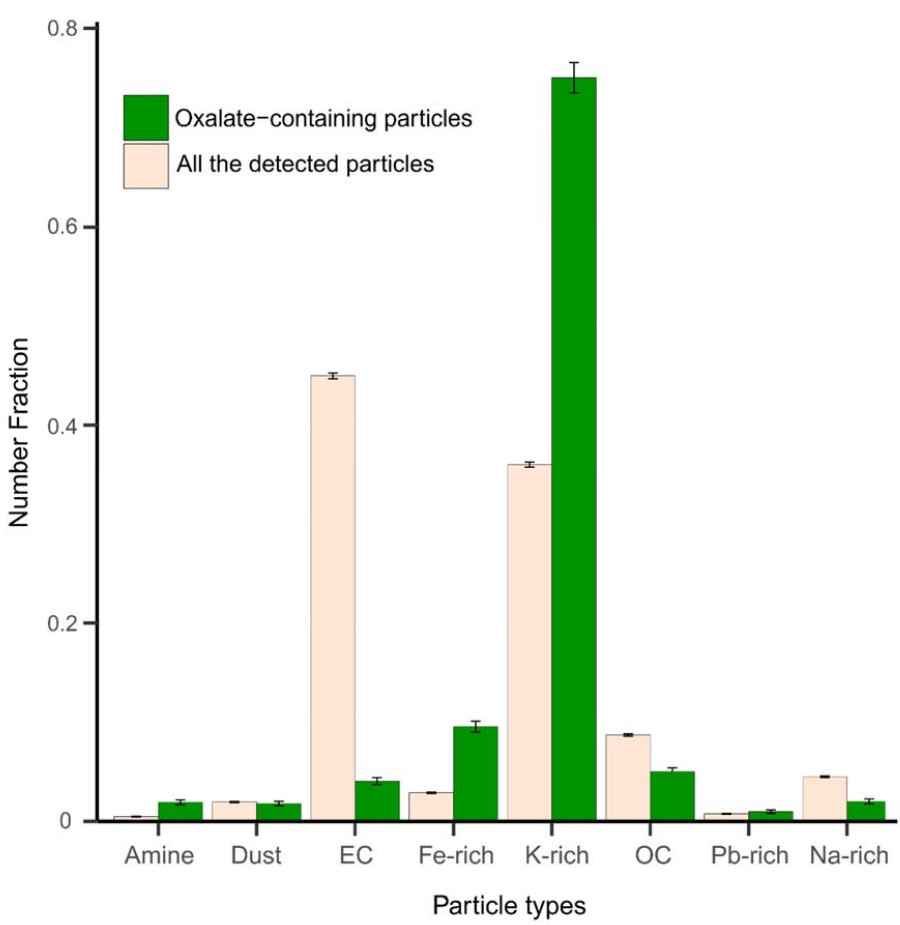


**Fig. 4.**



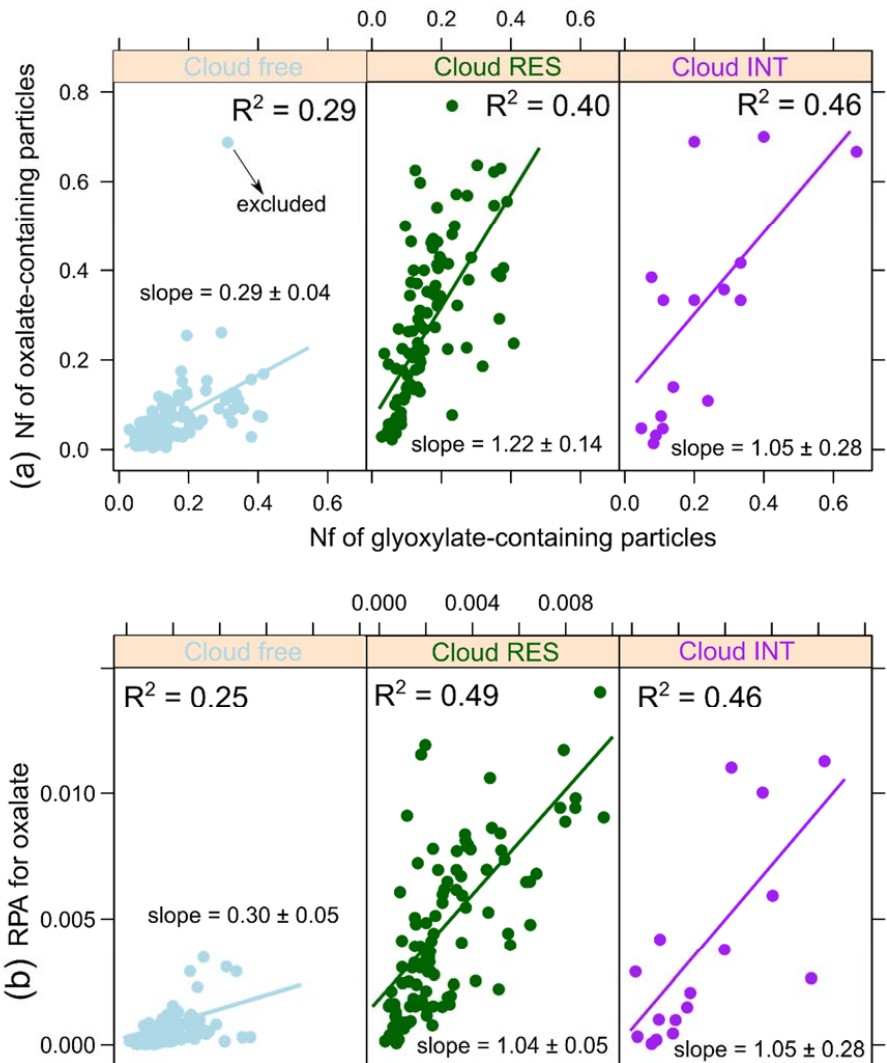


**Fig. 5.**



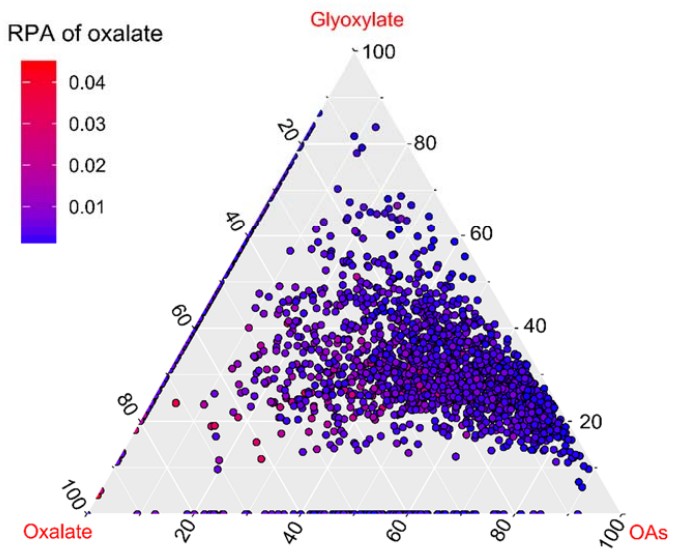

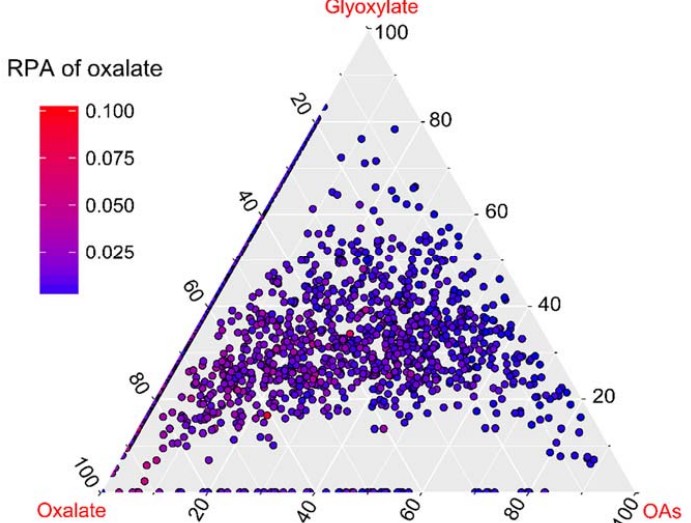


**Fig. 6.**





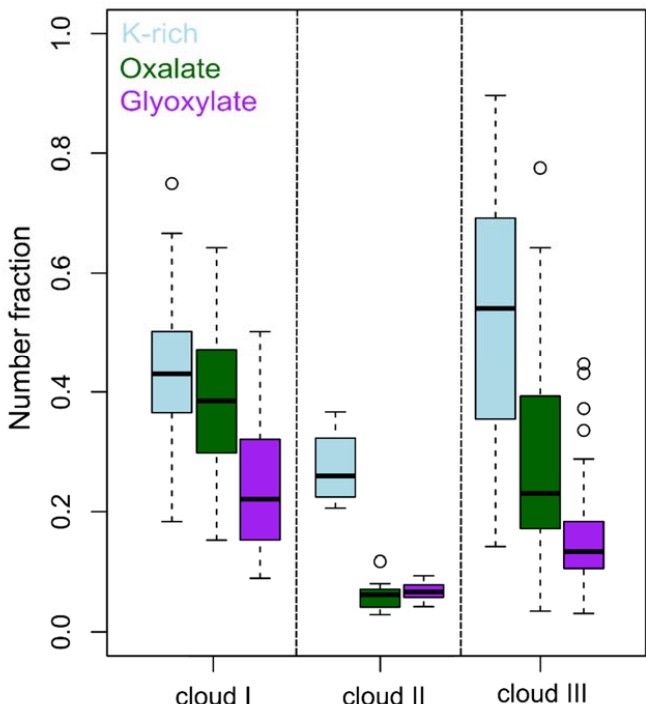


**Fig. 7.**