# Peer review of "Insight into the in-cloud formation of oxalate based on in situ measurement"

_Atmospheric Chemistry and Physics, 2017_

## Referee Comment (RC1) · Anonymous Referee #1 · 10 Sep 2017

The manuscript "Insight into the in-cloud formation of oxalate based on in situ measurement by single particle mass spectrometry" provides in-situ observations of oxalate-containing particles using single particle mass spectrometry combined with ground-based counterflow virtual impactor. This study aims to quantify single particle mixing sate and formation path of oxalate in cloud droplet residuals (cloud RES), the cloud interstitials (cloud INT), and ambient particles (PM 2.5) (cloud-free) at remote mountain site, in southern China during winter time in 2016. It concludes that oxalate-containing particles are highly correlated to aged biomass burning (potassium-rich) particles during the study period. In addition, enrichment of various organic acids in aged biomass burning particles is a control factor for oxalate formation. The results suggest that cloud

processing is the regional dominant formation mechanism for oxalate production with glyoxylate as a major intermediate.

The topic of this paper is relevant to the journal and has importance scientifically. The experiment design and data analysis presented are well done. However, in discussion section, the authors need to provide more sufficient in-depth scientific interpretation and discussion rather than make simple comparisons and draw conclusions in a rush by citing previous studies. Prior to publication, the authors should address the specific comments below.

1. Line 86: For the sake of completeness, authors can include the following study based on aircraft data:

Sorooshian, A., S. M. Murphy, S. Hersey, R. Bahreini, H. Jonsson, R. C. Flagan, and J. H. Seinfeld (2010). Constraining the contribution of organic acids and AMS m/z 44 to the organic aerosol budget: On the importance of meteorology, aerosol hygroscopicity, and region, Geophys. Res. Lett., 37, L21807, doi:10.1029/2010GL044951.

2. Line 99 and Line103: Shouldn't the appropriate reference be Sullivan and Prather, 2007 instead of Sullivan et al., 2007?

3. Line 134- 136: "The first one was a ground-based counterflow virtual impactor (GCVI) (Model 1205, Brechtel Mfg. Inc., USA), applied to collect the cloud RES particles with a diameter greater than 8$\mu$m."

Is the 8 micron in reference to droplets or the actual particle size? I believe it is the droplet size, and so more careful wording is required here to not say it is the particles that have diameters above 8 micron.

4. Suggest restructuring section 2.1 and 2.2 into one section, since currently section 2.2 doesn't provide many details about instrumentation for the study.

5. Line 183- Line186: Nfs of oxalate-containing particles for the three types shown in Fig. 1 (b) are number fractions relative to total oxalate-containing particles or total

detected particles?

6. Line 200-201: "Oxalate-containing particles had higher Nfs in the smaller cloud-free particles, indicative of primary emission or photochemical production followed by condensation (Zauscher et al., 2013)."

It likely would be more clear to indicate the specific particle size range as it is shown in the Fig. 2, instead of using the word "smaller".

In Fig 2., Nfs of cloud-free particles show two peak Nf values ($\sim$0.1) at the very first and last point. What are the interpretations for the peak at largest dva? Previous studies have shown oxalic acid found in aged sea salt and mineral dust particles in both sub- and super-micron size range.

7. Line 209- 217: Improvement of Fig. S4 is required in order to support the comparisons between cloud RES and cloud-free particle types.

The current figure shows the trends of Nfs and RPA for all particles instead of straightforward comparisons among the different particle types and especially, it is hard to distinguish Cloud RES and Cloud INT. Wind direction is not helpful to separate them, since the two types might have same air mass origins (i.e. cloud event III). Suggest using different markers to represent the three types in Fig. S4.

8. Line 229: all major ion peaks in Fig. 3 show higher Nfs in oxalate-containing particles than ones in all particles, except m/z18 (ammonium). It is necessary to add discussion here for the difference between m/z 18 and the rest of the species, since it is an important message delivered by Fig 3.

9. Line 234- 237: It is better to first introduce organic species names along with their possibly representative m/z Da at Line 230.

It is unclear how the correlation matrix (Table S1) can indicate the similar formation mechanism among the species. More interpretations are expected here.

10. Line 271 – 276: It is inappropriate to state Fe facilitated the formation of oxalate.

Sorooshian et al. (2013) has observed anti-correlation between Fe and Oxalate in cloud water vertical profiles over California coastal region. Similarly, oxalate is significantly lost through the photolysis of iron oxalato complexes as shown by the study over the rural area of PRD in China (Cheng et al., 2017). Thoughtful interpretations are required here.

11. Line 292- 294: For results of Nf, Cloud INT yields the highest R2 for instead of Cloud RES. Any interpretation for this? In addition, Fig. 5 shows identical linear fitting result for Nfs and RPA of Cloud INT particles, which is suspicious. Please check and confirm.

12. Line 302 -303: "To our knowledge, it is the first report on the direct link and the internally mixing state between glyoxylate and oxalate during in-cloud processing with high time resolution. "

The conclusion is not convincing by only using simple linear correlation analysis of Nf (and RPA) for glyoxylate- and oxalate-containing particles. Although, it indicates highest linear correlation is found between glyoxylate and oxalate, what are the results for the other major OAs shown in Fig. 3 and table S2? Do the authors also have significant (positive) correlations with oxalate-containing particles?

13. Line 313- 321: it is unclear what the (major) OAs referred to are, as they are not shown in Fig. S7. Plots are not labeled in the figure, and therefore it is difficult to follow the context here. Improvement is required.

14. Line 326- 327: "If this pathway dominated in this study, glyoxylate and oxalate should be evenly distributed in all the particle types..." It is unclear to have such expectation for glycoylate and oxalate based on previous context (Line 322 – Lin 326). Better interpretations are required.

Minor Comments

1. References should be cited in order of publication year from the oldest to the latest. Corrections are required through out the current manuscript (i.e. Line 58-59; Line 63; Line 67; Line 86-87; Line 324, etc.).

2. Line 86: appropriate preposition is "over" instead of "above"

3. Line 186: "Figure 1" should be "Fig. 1"

4. Line 190: typo, "a species", should be singular not plural.

5. Line 224: there is an extra single space between "39" and "Da".

6. In Fig. S5, plots on left don't have corresponding specie names with each m/z Da as ones on the right. Consistency is required.
* * *

---

## Referee Comment (RC2) · Anonymous Referee #2 · 30 Sep 2017

Review for "Insight into the in-cloud formation of oxalate based on in situ measurement by single particle mass spectrometry" by Zhang et al. submitted to ACPD

Overall comments

This paper presents investigation of in-cloud formation of oxalate based on single particle analysis of oxalate at a remote mountain site. Size-resolved mixing state of oxalate was analyzed separately in the cloud droplet residual (cloud RES), the cloud interstitial (cloud INT), and ambient (cloud-free) particles by single particle mass spectrometry. Several reasonable results were found including the enriched aged BB aerosol was mixed with oxalate and the enhanced formation of oxalate in the cloud RES and INT

particles. The investigation of the relationship between oxalate and organic acid ions also found glayoxylate as an important intermediate for the in cloud formation of oxalate. The topic is of great interest to a certain amount of readers and also proper for the scopes of the publication of this issue. However there are several general questions need to be answered before it can be considered for publication in this journal.

Comments

1. Definition: the determination of oxalate is not clear. It is according to the peak are or RPA of -89 larger than xxx? The definition of OA is also not clear. Since the manuscript refer to the calculation of OA intensity, the author should include the detail information in the section 2.3 or in the supporting informations.

2. Figure S7: Figure legend is not clear. No a-h is labeled, the label "cloud-free" is better on top of "cloud-RES", open circle shows all the data?

3. It also can be connected with the time cloud last as it can be clearly seen the second cloud event last less time and did not have such a high mixing ratio of oxalate compared with the other events. The second event is unique. Author can investigate a little bit on this issue.

4. 319-321 Author showed a statistics of OA for Cloud-free, RES and INC. The reviewer just curious how about the time series of these OA markers and oxalate in this campaign? Is there any good anti-trends?

5. 322-325 The definition of organic acid is also one critical issue. As we all known levoglucosan also have fragment peaks in -45, -59 and -73. Biomass burning particles have abundant levoglucosan and it will also decay in the atmosphere during the aerosol aging processes . Is there possible some of these ions are partially levoglucosan? More detail discussion should be added regarding to the diagnosis of these organic acid peaks.

6. Section 3.4 Line354: K-rich and oxalate showed really low R2 really surprised me.

Is that really fresh biomass burning aerosol in the cloud-free case?

7. Line 354-358 cloud water content is not discussed in this manuscript, is there possible that the cloud water content influenced the process, or other factors? Cloud can promote formation of oxalate but it can also scavenge water soluble ions (Zhou et al. 2010ïijŻ Wang et al. 2012). More discussion can be added.

Y. Zhou, T. Wang, X. Gao, L. Xue, X. Wang, Z. Wang, J. Gao, Q. Zhang, W. Wang.Continuous observations of water-soluble ions in PM2.5 at Mount Tai (1534 m a.s.l.) in central-eastern China, Journal of Atmospheric Chemistry, 2010, 64, 107-127
Z. Wang, T. Wang, J. Guo, R. Gao, L. Xue, J. Zhang, Y. Zhou, X. Zhou, Q. Zhang, W. Wang, Formation of secondary organic carbon and cloud impact on carbonaceous aerosols at Mount Tai, North China. Atmospheric Environment, 2012, 46, 516-527

8. Figure 2. It is quite ingesting that the cloud-free oxalate showed a peak with such a small size. If the data is correct, it might be fresh emitted biomass burning aerosols. Is there any other evidence to support this phenomena?

Others

1. Figure 1, Cloud-free is better on top of other two labels.

2. Figure 3, color coded digital data with peak area information of oxalate can include more information.

3. Figure 5, regression method should be included. The author can refer the software made by Wu et al. 2017

https://www.atmos-meas-tech-discuss.net/amt-2017-300/

4. Table S1, K.rich should be K-rich; Table S2 what's 41 for? Table S3 what's 45 for?

5. Table S3 normalized by ???

[Figure]

2017.

---

## Author Response (AR1)

**Response to comments**

**Anonymous Referee #1**

*The manuscript "Insight into the in-cloud formation of oxalate based on in situ measurement by single particle mass spectrometry" provides in-situ observations of oxalate containing particles using single particle mass spectrometry combined with ground based counterflow virtual impactor. This study aims to quantify single particle mixing sate and formation path of oxalate in cloud droplet residuals (cloud RES), the cloud interstitials (cloud INT), and ambient particles (PM 2.5) (cloud-free) at remote mountain site, in southern China during winter time in 2016. It concludes that oxalate-containing particles are highly correlated to aged biomass burning (potassium-rich) particles during the study period. In addition, enrichment of various organic acids in aged biomass burning particles is a control factor for oxalate formation. The results suggest that cloud processing is the regional dominant formation mechanism for oxalate production with glyoxylate as a major intermediate.*

*The topic of this paper is relevant to the journal and has importance scientifically. The experiment design and data analysis presented are well done. However, in discussion section, the authors need to provide more sufficient in-depth scientific interpretation and discussion rather than make simple comparisons and draw conclusions in a rush by citing previous studies. Prior to publication, the authors should address the specific comments below.*

We would like to thank the reviewer for his/her useful comments and recommendations to improve the manuscript. We have addressed the specific comments in the following text.

*1. Line 86: For the sake of completeness, authors can include the following study based on aircraft data:*
*Sorooshian, A., S. M. Murphy, S. Hersey, R. Bahreini, H. Jonsson, R. C. Flagan, and J. H.*

*Seinfeld (2010). Constraining the contribution of organic acids and AMS m/z 44 to the organic aerosol budget: On the importance of meteorology, aerosol hygroscopicity, and region, Geophys. Res. Lett., 37, L21807, doi:10.1029/2010GL044951.*

Thanks for the suggestion. We have included the citation (Sorooshian et al., 2010) as suggested.

*2. Line 99 and Line103: Shouldn't the appropriate reference be Sullivan and Prather, 2007 instead of Sullivan et al., 2007?*

Thanks for the suggestion. We have cited (Sullivan and Prather, 2007) instead of (Sullivan et al., 2007) as suggested.

*3. Line 134- 136: "The first one was a ground-based counterflow virtual impactor (GCVI) (Model 1205, Brechtel Mfg. Inc., USA), applied to collect the cloud RES particles with a diameter greater than 8 μm." Is the 8 micron in reference to droplets or the actual particle size? I believe it is the droplet size, and so more careful wording is required here to not say it is the particles that have diameters above 8 micron.*

We agree with the comment. We have corrected the sentence to "The first one was a ground-based counterflow virtual impactor (GCVI) (Model 1205, Brechtel Mfg. Inc., USA), applied to obtain the cloud RES particles from the cloud droplets larger than 8 μm.". Please refer to Lines 133-135 of the revised manuscript.

*4. Suggest restructuring section 2.1 and 2.2 into one section, since currently section 2.2 doesn't provide many details about instrumentation for the study.*

Thanks for the suggestion. Section 2.1 and 2.2 have been restructured into one section accordingly.

*5. Line 183- Line186: Nfs of oxalate-containing particles for the three types shown in Fig.*

*1 (b) are number fractions relative to total oxalate-containing particles or total detected particles?*

Nfs of oxalate-containing particles for the three types shown in Fig. 1 (b) are number fractions relative to the total detected particles. We have revised the figure caption to "Fig. 1. (a) Temporal variation (in one-hour resolution) of Nfs of the oxalate-containing particles, and box-and-whisker plots of (b) the Nfs of oxalate-containing particles as shown in (a)" to make it clear. Please refer to Lines 632-635 in the revised manuscript.

*6. Line 200-201: "Oxalate-containing particles had higher Nfs in the smaller cloud free particles, indicative of primary emission or photochemical production followed by condensation (Zauscher et al., 2013)." It likely would be more clear to indicate the specific particle size range as it is shown in the Fig. 2, instead of using the word "smaller". In Fig 2., Nfs of cloud-free particles show two peak Nf values (0.1) at the very first and last point. What are the interpretations for the peak at largest dva? Previous studies have shown oxalic acid found in aged sea salt and mineral dust particles in both sub and super-micron size range.*

Thanks for the comment. The sentence has been revised to "Oxalate-containing particles had higher Nfs in the cloud-free particles with $d_{va} < 0.4$ μm, indicative of primary emission or photochemical production followed by condensation (Zauscher et al., 2013). ".

We have checked the distribution of each particle type of the cloud-free oxalate-containing particles along $d_{va}$. The result shows that the oxalate-containing particles at the largest $d_{va}$ (1.3-1.4 μm) they were most likely contributed by the aged biomass burning particles, as shown below. However, it shouldn't be conclusive since only 12 particles were found at this size range. Please refer to Fig. S2 in the revised manuscript.

[Figure]

Fig S2. The size-resolved number fraction for each particle types of oxalate-containing particles.

*7. Line 209- 217: Improvement of Fig. S4 is required in order to support the comparisons between cloud RES and cloud-free particle types. The current figure shows the trends of Nfs and RPA for all particles instead of straightforward comparisons among the different particle types and especially, it is hard to distinguish Cloud RES and Cloud INT. Wind direction is not helpful to separate them, since the two types might have same air mass origins (i.e. cloud event III). Suggest using different markers to represent the three types in Fig. S4.*

Thanks for the comment. We have revised the Fig. S4 (shown as below) by using different markers to represent the three types in addition to the wind direction, in order to support the comparison between cloud RES, cloud INT, and cloud-free particles.

[Figure]

Figure S4. Scattering plots of (upper) the number fraction and (lower) the RPA of the oxalate-containing particles versus relative humidity, separated for the cloud-free, cloud RES, and cloud INT particles. The coloration indicates the wind direction.

*8. Line 229: all major ion peaks in Fig. 3 show higher Nfs in oxalate-containing particles than ones in all particles, except m/z 18 (ammonium). It is necessary to add discussion here for the difference between m/z 18 and the rest of the species, since it is an important message delivered by Fig 3.*

Thanks for the comment. We have analyzed the Nfs of ammonium associated with different particle types in oxalate-containing particles. The result indicates that the higher Nf of ammonium in all the detected particles rather than in the oxalate-containing particles is due to uneven distribution of ammonium among the different particle types. As can be seen in Fig. 4, oxalate was dominantly distributed in the K-rich particle type, which contained lower fraction of ammonium (~40%). However, as the dominant type in all the detected particles, EC type contained higher fraction (~80%) of ammonium. Therefore, the alkali nature (larger abundance of potassium, sodium) of the K-rich might explain the lower fraction of ammonium associated with the oxalate-containing particles. The discussion on this issue has been added in Lines 254-257 of the revised manuscript and Lines 72-79 of the revised Supplement.

*9. Line 234- 237: It is better to first introduce organic species names along with their possibly representative m/z Da at Line 230. It is unclear how the correlation matrix (Table S1) can indicate the similar formation mechanism among the species. More interpretations are expected here.*

Thanks for the suggestion. We have introduced the names of organic species with possibly representative m/a Da in Lines 233-235 of the revised manuscript. Meanwhile, we have revised the sentences to "Their RPAs increased with increasing particle sizes (Fig. S5), indicative of secondary origins (Zauscher et al., 2013). In addition, their Nfs tracked each other temporally in cloud-free particles (Table S1), supporting their similar formation mechanisms, most likely formed through photochemical oxidation followed by gas-to-particle partition (Zauscher et al., 2013).", to indicate the similar formation mechanism among the species. Please refer to Lines 233-238 of the revised manuscript.

*10. Line 271 – 276: It is inappropriate to state Fe facilitated the formation of oxalate. Sorooshian et al. (2013) has observed anti-correlation between Fe and Oxalate in cloud water vertical profiles over California coastal region. Similarly, oxalate is significantly lost through the photolysis of iron oxalato complexes as shown by the study over the rural area of PRD in China (Cheng et al., 2017). Thoughtful interpretations are required here.*

We agree with the comment. Iron might play an important role in the sink of oxalate. However, it is unlikely to be an important factor in this study, mostly with the occurrence of orographic cloud and also possibly low radiation in winter. Therefore, it is different from the observation by Sorooshian et al. (2013) and Cheng et al. (2007), which was likely associated with high radiation. We have moved the discussion to the Supplement and added some interpretations as follows: "As shown in Fig. 4, ~10% of oxalate was associated with Fe-rich particles, second only to the K-rich particles. Regarding that the Fe-rich particles only accounted for $2.5 \pm 0.4\%$ of all the detected particles (Lin et al., 2017), it might reflect that the Fe facilitated the formation of oxalate. Fenton reactions involving iron can produced more oxidants (e.g., •OH) (Nguyen et al., 2013; Herrmann et al., 2015), which would enhanced the formation of oxalate (Ervens et al., 2014). While Sorooshian et al. (2013), Zhou et al. (2015), and Cheng et al. (2017) have suggested that oxalate can be significantly lost through the photolysis of iron-oxalato complexes. The difference between these observations and this study might be attributed to the different radiation. Our observation was conducted at a mountain site in winter, mostly covered with orographic cloud, resulted in very low visibility (< 500 m), and thus low radiation was expected during sampling. With sampling conducted on an aircraft, cloud water collected by Sorooshian et al. (2013)

included the below and top of cloud water samples, and thus photolysis is expected. On the other hand, the highest fraction (> 30%) of oxalate was found to be internally mixed with metal-containing (e.g., iron, zinc, copper) particles in the Pearl River Delta region (Cheng et al., 2017). The internally mixed oxalate and iron could account for ~50% of iron particles at nighttime (Zhou et al., 2015). Additionally, oxalate was also found to be slightly enriched in amine-containing particles, which is most probably attributed to the enhanced partition of amine to wet aerosols (Rehbein et al., 2011; Zhang et al., 2012).". Please refer to the revised Supplement.

*11. Line 292- 294: For results of Nf, Cloud INT yields the highest R2 for instead of Cloud RES. Any interpretation for this? In addition, Fig. 5 shows identical linear fitting result for Nfs and RPA of Cloud INT particles, which is suspicious. Please check and confirm.*

Thanks for the comment. We have checked the data and confirmed the results shown in the Fig. 5. The highest $R^2$ of Nf for cloud INT particles is explained by the number of samples applied in the analysis, which is significantly less for cloud INT particles ($N =$ 16 for cloud INT particles, $N = 107$ for cloud RES particles). $R^2$ in the analysis is defined as the square of the correlation between the response values and the predicted response values. Therefore, it might be inappropriate to make a comparison between $R^2$ for these distinctly different samples. It is also noted that statistical hypothesis testing shows that the *p*-value is $1.7*10^{-13}$ and 0.002 for cloud RES and INT particles, respectively. The sample number used in the analysis has been added in the caption of Fig. 5 (Line 648 of the revised manuscript) to make it clear.

*12. Line 302 -303: "To our knowledge, it is the first report on the direct link and the internally mixing state between glyoxylate and oxalate during in-cloud processing with high time resolution." The conclusion is not convincing by only using simple linear correlation analysis of Nf (and RPA) for glyoxylate- and oxalate-containing particles. Although, it indicates highest linear correlation is found between glyoxylate*

*and oxalate, what are the results for the other major OAs shown in Fig. 3 and table S2? Do the others also have significant (positive) correlations with oxalate-containing particles?*

Thanks for the comment. In addition to the linear correlation analysis between glyoxylate- and oxalate-containing particles in the Nf and RPA in Fig. 5, we have also shown in Fig. 3 that more than half of oxalate-containing particles contained glyoxylate, in order to confirm the direct link between glyoxylate and oxalate. Besides, oxalate also shows significant correlation ($p < 0.001$) with other OAs as shown in Table S2. However, we only analyzed in detail the relationship between glyoxylate and oxalate in this manuscript, since glyoxylate is an important intermediate for the formation of oxalate, which is confirmed by the highest correlation between them, and the analysis shown in section 3.3. Please refer to Table S2 and section 3.3 of the revised manuscript.

*13. Line 313- 321: it is unclear what the (major) OAs referred to are, as they are not shown in Fig. S7. Plots are not labeled in the figure, and therefore it is difficult to follow the context here. Improvement is required.*

Thanks for the comment. We have corrected the Fig. S7 to make it clear. Fig. S7 is shown as followed in the revised Supplement.

[Figure]

Fig. S7. Box and whisker plot of the variations of number fractions for four OAs in (a-d) all the detected particles, and (e-h) oxalate-containing particles, separated for cloud-free, RES, and INT particles, respectively.

*14. Line 326- 327: "If this pathway dominated in this study, glyoxylate and oxalate should be evenly distributed in all the particle types: : :" It is unclear to have such expectation for glyoxylate and oxalate based on previous context (Line 322 – Lin 326). Better interpretations are required.*

Thanks for the comment. We have revised the sentence to "Assuming that the in-cloud formation of oxalate was dominantly contributed from the volatile organic compounds, glyoxylate and oxalate would be evenly formed in all the particle types," to make it clear. We also explain in the following text that "This is inconsistent with our observation that oxalate was predominantly associated with the aged biomass burning particles (Fig. 3). It indicates that a certain amount of glyoxylate should be directly produced in cloud from the organics formed before the cloud events and associated with aged biomass burning particles.". Please refer to Lines 325-330 of the revised manuscript.

*Minor Comments*
*1. References should be cited in order of publication year from the oldest to the latest. Corrections are required through out the current manuscript (i.e. Line 58-59; Line 63; Line 67; Line 86-87; Line 324, etc.).*

They have been corrected accordingly.

*2. Line 86: appropriate preposition is "over" instead of "above"*

It has been corrected accordingly.

*3. Line 186: "Figure 1" should be "Fig. 1"*

It has been corrected accordingly.

*4. Line 190: typo, "a species", should be singular not plural.*

It has been corrected accordingly.

*5. Line 224: there is an extra single space between "39" and "Da".*

It has been corrected accordingly.

*6. In Fig. S5, plots on left don't have corresponding specie names with each m/z Da as ones on the right. Consistency is required.*

We have added the corresponding specie names with each m/z Da in the figure as suggested.

**Response to comments**

**Anonymous Referee #2**

*Review for "Insight into the in-cloud formation of oxalate based on in situ measurement by single particle mass spectrometry" by Zhang et al. submitted to ACPD*

*Overall comments:*

*This paper presents investigation of in-cloud formation of oxalate based on single particle analysis of oxalate at a remote mountain site. Size-resolved mixing state of oxalate was analyzed separately in the cloud droplet residual (cloud RES), the cloud interstitial (cloud INT), and ambient (cloud-free) particles by single particle mass spectrometry. Several reasonable results were found including the enriched aged BB aerosol was mixed with oxalate and the enhanced formation of oxalate in the cloud RES and INT particles. The investigation of the relationship between oxalate and organic acid ions also found glayoxylate as an important intermediate for the in cloud formation of oxalate. The topic is of great interest to a certain amount of readers and also proper for the scopes of the publication of this issue. However there are several general questions need to be answered before it can be considered for publication in this journal.*

We would like to thank the reviewer for his/her useful comments and recommendations to improve the manuscript. We have addressed the specific comments in the following text.

Comments

1. *Definition: the determination of oxalate is not clear. It is according to the peak are or RPA of -89 larger than xxx? The definition of OA is also not clear. Since the*

*manuscript refer to the calculation of OA intensity, the author should include the detail information in the section 2.3 or in the supporting information.*

Thanks for the suggestion. We have added "The identified ion peaks have peak areas larger than 5 (arbitrary unit), whereas the noise level is lower than 1." in Lines 29-30 of the revised supplement to make it clear.

2. *Figure S7: Figure legend is not clear. No a-h is labeled, the label "cloud-free" is better on top of "cloud-RES", open circle shows all the data?*

Thanks for the comments. We have revised the Figure S7 as suggested. Open circles shows the data not included between the whiskers, which is larger than 90 percentiles or lower than 10 percentiles of the data set. Please refer to the caption of Figure S7 in the Supplement.

3. *It also can be connected with the time cloud last as it can be clearly seen the second cloud event last less time and did not have such a high mixing ratio of oxalate compared with the other events. The second event is unique. Author can investigate a little bit on this issue.*

We agree with the comment that the second event is unique. As we stated in the manuscript, air mass analysis showed that cloud II was strongly influenced by northeastern air mass, contrasting to the southwestern air mass during cloud I and III (Lin et al., 2017). However, short cloud processing time cloud not be the reason for the lower Nf of oxalate-containing particles during cloud II. As can be seen in Fig. 1, the Nf of oxalate-containing particles increased to 20% within several hours during cloud I and III. Therefore, we indicated that in-cloud production of oxalate on the aged biomass burning particles is dominantly controlled by the glyoxylate, which substantially decreased during cloud II, relative to Cloud I and III. As also suggested by the reviewer, we added some discussion on the cloud water content, as "Cloud water content plays an important role in both the formation and scavenging of water soluble ions (Zhou et al., 2009; Wang et al., 2012), and thus might contribute to the lower fraction of oxalate during cloud II. Model simulation indicates that the formation of oxalate is as a function of cloud processing time and droplet sizes, which directly links to the cloud water content (Sorooshian et al., 2013). With visibility as an indicator (Table S3), it shows the lowest cloud water content during cloud II. However, non-significant correlation was found between the Nf of the oxalate-containing particles and visibility.". Please refer to section 3.4 of the revised manuscript.

4. *319-321 Author showed a statistics of OA for Cloud-free, RES and INC. The reviewer just curious how about the time series of these OA markers and oxalate in this campaign? Is there any good anti-trends?*

Thanks for the comments. It might be expected anti-trends between oxalate and OA markers when the total amount of OAs is constant in a close system. However, in open air, it might be expected positive correlations as analyzed in Table S1. It is reasonable since these OAs served as important precursors for the formation of oxalate. The statistics in Fig. S7 supports the conversion of OAs to oxalate via showing the decrease of the Nfs of OAs associated with the oxalate-containing particles. This only provides evidence for the conversion of OAs to oxalate during cloud events.

5. *322-325 The definition of organic acid is also one critical issue. As we all known levoglucosan also have fragment peaks in -45, -59 and -73. Biomass burning particles have abundant levoglucosan and it will also decay in the atmosphere during the aerosol aging processes. Is there possible some of these ions are partially levoglucosan? More detail discussion should be added regarding to the diagnosis of these organic acid peaks.*

We agree with the comment that levoglucosan from biomass burning also have fragment peaks in m/z -45, -59 and -73. Thus, it is also possible that some of these ions are partly from levoglucosan. However, these ion peaks were most likely from secondary species in the present study, as discussed in the revised manuscript. This is probably explained by that their RPAs increased with increasing particle diameters (Fig. S5), consistent with that observed by Zauscher et al (2013). We indicate that these organics, most likely assigned to be formate at m/z $-45[HCO_2]-$, acetate at m/z $-59[CH_3CO_2]-$, methylglyoxal or acrylate at m/z $-71[C_2H_3CO_2]-$, and glyoxylate at m/z $-73[C_2HO_3]-$ (Zauscher et al., 2013). In addition, their Nfs tracked each other temporally in cloud-free particles (Table S1), supporting their similar formation mechanisms, most likely formed through photochemical oxidation followed by gas-to-particle partition (Zauscher et al., 2013). Please refer to Lines 233-242 of the revised manuscript.

6. *Section 3.4 Line354: K-rich and oxalate showed really low $R^2$ really surprised me. Is that really fresh biomass burning aerosol in the cloud-free case?*

Thanks for the comment. In this study, the K-rich particles were highly aged, and heavily internally mixed with sulfate and nitrate (Lin et al., 2017). As analyzed in section 3.4, it is shown the higher correlation between the Nfs of oxalate-containing and glyoxylate-containing particles, relative to that between the Nfs of oxalate-containing particles and the aged biomass burning particles (Table S1). The result suggests that the formation of oxalate is more dependent on the amount of glyoxylate rather than the amount of biomass burning aerosol, which might be influenced by the burning condition and meteorological conditions during the transport.

Lin, Q., Zhang, G., Peng, L., Bi, X., Wang, X., Brechtel, F. J., Li, M., Chen, D., Peng, P., Sheng, G., and Zhou, Z.: In situ chemical composition measurement of individual cloud residue particles at a mountain site, southern China, Atmos. Chem. Phys., 17, 8473-8488, doi:10.5194/acp-17-8473-2017, 2017.

7. *Line 354-358 cloud water content is not discussed in this manuscript, is there*

*possible that the cloud water content influenced the process, or other factors? Cloud can promote formation of oxalate but it can also scavenge water soluble ions (Zhou et al. 2010; Wang et al. 2012). More discussion can be added.*

*Y. Zhou, T. Wang, X. Gao, L. Xue, X. Wang, Z. Wang, J. Gao, Q. Zhang, W. Wang. Continuous observations of water-soluble ions in PM2.5 at Mount Tai (1534 m a.s.l.) in central-eastern China, Journal of Atmospheric Chemistry, 2010, 64, 107-127*

*Z. Wang, T. Wang, J. Guo, R. Gao, L. Xue, J. Zhang, Y. Zhou, X. Zhou, Q. Zhang, W. Wang, Formation of secondary organic carbon and cloud impact on carbonaceous aerosols at Mount Tai, North China. Atmospheric Environment, 2012, 46, 516-527*

We agree with the comment that cloud water content might be an important factor that influences the oxalate formation in the droplets. Such discussion has been added in this section as "Cloud water content plays an important role in both the formation and scavenging of water soluble ions (Zhou et al., 2009; Wang et al., 2012), and thus might contribute to the lower fraction of oxalate during cloud II. Model simulation indicates that the formation of oxalate is as a function of cloud processing time and droplet sizes, which directly link to the cloud water content (Sorooshian et al., 2013). With visibility as an indicator (Table S3), it shows the lowest cloud water content during cloud II. However, non-significant correlation was found between the Nf of the oxalate-containing particles and visibility.".

*8. Figure 2. It is quite ingesting that the cloud-free oxalate showed a peak with such a small size. If the data is correct, it might be fresh emitted biomass burning aerosols. Is there any other evidence to support this phenomena?*

We agree with the comment that the peak at such a small size might be contributed from the freshly emitted biomass burning aerosols. However, this peak is most likely attributed to the photochemical production, since these smaller particles (0.1 - 0.3 μm) were extensively (nearly 100%) internally mixed with secondary species, such as sulfate and nitrate. The discussion has been included in Lines 202-204 of the revised manuscript.

Others

1. *Figure 1, Cloud-free is better on top of other two labels.*

It has been revised as suggested.

2. *Figure 3, color coded digital data with peak area information of oxalate can include more information.*

Figure 3 was shown to illustrate the predominant association of the major OAs with the oxalate-containing particles, rather than all the detected particles. We agree with the comment that peak area information of oxalate would provide some useful information. Actually, this information is shown in Fig. S1, and thus we only showed the number fraction in Fig. 3 for simplicity. In addition, we also compared the variation of the peak area distribution of oxalate, glyoxylate, and the major OAs in Fig. 6 to investigate the transformation of OAs to oxalate. Please refer to Fig. S1 and Fig. 6 of the revised manuscript.

3. *Figure 5, regression method should be included. The author can refer the software made by Wu et al. 2017 https://www.atmos-meas-tech-discuss.net/ amt-2017-300/*

Thanks for the suggestion. We have included the method. The caption of Fig. 5 has been revised to "Simple linear regression (with least-square method) between (a) the Nfs and (b) The RPAs of the oxalate-containing and glyoxylate-containing particles, separated for the cloud-free, cloud RES, and cloud INT particles, respectively.". The least-square approach is applied in this work, although Wu et al. (2017) recommended other regression methods (such as DR, WODR and YR). It is because (1) simple linear regression with F-test allows for the testing on the correlation of our data, (2) we did not attempt to quantify the slope from the analysis, and (3) the recommended regression methods need appropriate weighting, which would complicate the analysis for the single particle data since it is hard to provide appropriate uncertainties.

4. T*able S1, K.rich should be K-rich; Table S2 what's 41 for? Table S3 what's 45 for?*

We have corrected the mistake in Table S1. In Table S2 and S3, the ion peaks at m/z -45, -59, -71, and -73 stands for formate, acetate, methylglyoxal or acrylate, and glyoxylate, respectively. Please refer to Lines 233-235 of the revised manuscript.

5. *Table S3 normalized by ???*

We did not normalized the data in Table S3. We only showed the normalized data in Fig. S3, which is normalized by the largest number over the size bins.

[revised manuscript text omitted]

**(a) cloud-free oxalate-containing particles**

[Figure]

**(b) cloud RES and INT oxalate-containing particles**

[Figure]

**Fig. 6.**

[Figure]

**Fig. 7.**

**Insight into the in-cloud formation of oxalate based on in-situ measurement by single**

**particle mass spectrometry**

Guohua Zhang[1], Qinhao Lin[1,2], Long Peng[1,2], Yuxiang Yang[1,2], Yuzhen Fu[1,2], Xinhui Bi[1,*],

Mei Li[3], Duohong Chen[4], Jianxin Chen[5], Zhang Cai[6], Xinming Wang[1], Ping'an Peng[1],

Guoying Sheng[1], Zhen Zhou[3]

[1] State Key Laboratory of Organic Geochemistry and Guangdong Key Laboratory of

Environmental Resources Utilization and Protection, Guangzhou Institute of Geochemistry,

Chinese Academy of Sciences, Guangzhou 510640, PR China

[2] University of Chinese Academy of Sciences, Beijing 100039, PR China

[3] Institute of Mass Spectrometer and Atmospheric Environment, Jinan University,

Guangzhou 510632, China

[4] State Environmental Protection Key Laboratory of Regional Air Quality Monitoring,

Guangdong Environmental Monitoring Center, Guangzhou 510308, PR China

[5] Shaoguan Environmental Monitoring Center, Shaoguan 512026, PR China

[6] John and Willie Leone Family Department of Energy and Mineral Engineering, The

Pennsylvania State University, University Park, PA 16802, USA

**Instrumentation**

**SPAMS**

Individual particles are introduced into SPAMS through a critical orifice. They are focused and accelerated to specific velocities, which are determined by two continuous diode Nd:YAG laser beams (532 nm). Based on the measured velocities, a pulsed laser (266 nm) downstream is trigger to desorp/ionize the particles. The produced positive and negative molecular fragments are recorded. In summary, a velocity, a detection moment, and an ion mass spectrum are recorded for each ionized particle, while there is no mass spectrum for not ionized particles. The velocity could be converted to $d_{va}$ based on a calibration using polystyrene latex spheres (PSL, Duke Scientific Corp., Palo Alto) with predefined sizes. The identified ion peaks have peak areas larger than 5 (arbitrary unit), whereas the noise level is lower than 1.

**An discussion on the preferential association of oxalate within Fe-rich and Amine**

**particles**

As shown in Fig. 4, ~10% of oxalate was associated with Fe-rich particles, second only to the K-rich particles. Regarding that the Fe-rich particles only accounted for 2.5 ±

0.4% of all the detected particles (Lin et al., 2017), it might reflect that the Fe facilitated the formation of oxalate. Fenton reactions involving iron can produced more oxidants (e.g.,

•OH) (Nguyen et al., 2013; Herrmann et al., 2015), which is an important factor for the formation of oxalate (Ervens et al., 2014). While Sorooshian et al. (2013), Zhou et al.

(2015), and Cheng et al. (2017) have suggested that oxalate can be significantly lost through the photolysis of iron-oxalato complexes. The difference between these observations and this study might be attributed to the different radiation. Our observation was conducted at a mountain site in winter, mostly covered with orographic cloud, resulted in very low visibility (< 500 m), and thus low radiation was expected during sampling. With sampling conducted on an aircraft, cloud water collected by Sorooshian et al. (2013) included the below and top of cloud water samples, and thus photolysis is expected. On the other hand, the highest fraction (> 30%) of oxalate was found to be internally mixed with metal-containing (e.g., iron, zinc, copper) particles in the Pearl River Delta region (Cheng et al., 2017). The internally mixed oxalate and iron could account for ~50% of iron particles at nighttime (Zhou et al., 2015). Additionally, oxalate was also found to be slightly enriched in amine-containing particles, which is most probably attributed to the enhanced partition of amine to wet aerosols (Rehbein et al., 2011; Zhang et al., 2012).

Table S1. Correlation analysis between the hourly detected number for species in cloud-free particles (N = 109) and ***cloud RES*** particles (N = 123). Most of the analysis shows significant correlation ($p < 0.001$) between the species, with the $R^2$ shown as follows.

Results without significant correlation are marked with superscripts a and b.

|  | m/z -45 | m/z -59 | m/z -71 | m/z -73 | m/z -89 | K-rich |
|---|---|---|---|---|---|---|
| **m/z -45** | 1 | | | | | |
| **m/z -59** | 0.92/*0.93* | 1 | | | | |
| **m/z -71** | 0.77/*0.33* | 0.92/*0.35* | 1 | | | |
| **m/z -73** | 0.94/*0.81* | 0.92/*0.86* | 0.80/*0.20* | 1 | | |
| **m/z -89** | 0.22/*0.32* | 0.38/*0.45* | 0.46/*0.12* | 0.33/*0.64* | 1 | |
| **K-rich** | 0.52/**0.58** | 0.33/**0.59** | 0.21/0 [a] | 0.57/*0.72* | 0.05 [b] /**0.41** | 1 |

[a] $p = 0.37$; [b] $p = 0.009$.

Table S2. Number fraction (%) of ion peaks for organic acids associated with all the detected particles and K-rich particles, respectively.

| Ion peaks | All the detected particles (%) | K-rich particles (%) |
|---|---|---|
| m/z -45 | 12.4 ± 0.1 | 21.5 ± 0.3 |
| m/z -59 | 8.6 ± 0.1 | 16.5 ± 0.3 |
| m/z -71 | 2.8 ± 0.1 | 5.6 ± 0.1 |
| m/z -73 | 12.6 ± 0.1 | 22.5 ± 0.3 |

Table S3. Number fraction (%) of the major OAs relative to all the detected particles, and visibility during each cloud event. Visibility was used here to indicate the cloud water content, since visibility is mainly controlled by the droplet number in cloud.

| Ion peaks | Cloud I | Cloud II | Cloud III |
|---|---|---|---|
| m/z -45 | 16.5 ± 11.1 | 4.8 ± 1.2 | 8.6 ± 4.7 |
| m/z -59 | 16.0 ± 9.6 | 3.9 ± 1.2 | 8.6 ± 5.5 |
| m/z -71 | 8.7 ± 7.3 | 0.6 ± 0.4 | 4.0 ± 4.1 |
| Visibility (km) | 0.05 ± 0.03 | 0.31 ± 0.69 | 0.11 ± 0.17 |

[Figure]

**Figure S1.** The number-based digitized mass spectrum of all the detected oxalate-containing particles. Compared with the number fraction of ammonium in Fig. 3, the result shows higher Nfs in oxalate-containing particles than ones in all particles, except m/z 18 (ammonium). As can be seen in Fig. 4, oxalate was dominantly distributed in K-rich particle type, which contained lower fraction of ammonium (~40%). However, as the dominant type in all the detected particles, EC type contained higher fraction (~80%) of ammonium. Therefore, the alkali nature (larger abundance of potassium, sodium) of the K-rich might explain the lower fraction of ammonium associated with the oxalate-containing particles.

(a)

[Figure]

(b)

[Figure]

Figure S2. (a) Average mass spectra and (b) the size-resolved number fraction for each particle type of oxalate-containing particles. Representative ions peaks were labeled for each particle types. One may expect that oxalate at the largest $d_{va}$ (1.3-1.4 μm) is associated with aged sea salt and/or mineral dust particles. However, our result shows that the aged biomass burning particles could contribute to the largest $d_{va}$ (1.3-1.4 μm)

mode of oxalate. However, it shouldn't be conclusive since only 12 particles were found at this size range.

[Figure]

Figure S3. The normalized unscaled number size distribution of oxalate-containing particles in cloud-free, RES, and INT particles, respectively.

[Figure]

Figure S4. Scattering plots of (upper) the number fraction and (lower) the RPA of the oxalate-containing particles versus relative humidity, separated for the cloud-free, cloud

RES, and cloud INT particles . The coloration indicates the wind direction.

[Figure]

Figure S5. Size-resolved distribution of RPAs for each species in the cloud-free and RES

particles.

[Figure]

Figure S6. Number fraction of each oxalate-containing particle type for the (a) cloud-free, (b) cloud RES, and (c) cloud INT particles, respectively.

[Figure]

Figure S7. Box and whisker plot of the variations of number fractions for four OAs in (a- d) all the detected particles, and (e-h) oxalate-containing particles, separated for cloud-free,

RES, and INT particles, respectively. In a box and whisker plot, the lower, median and upper line of the box denote the 25, 50, and 75 percentiles, respectively; the lower and upper edges of the whisker denote the 10 and 90 percentiles, respectively. Open circles shows the data not included between the whiskers, which is larger than 90 percentiles or lower than 10 percentiles of the data set.